# Recombinant Plasminogen Activator of the Sandworm (*Perinereis aibuhitensis*) Expression in *Escherichia coli*

**DOI:** 10.3390/bioengineering11101030

**Published:** 2024-10-15

**Authors:** Tuo Song, Xiaozhen Diao, Jun Cheng, Yang Man, Boyu Chen, Haixing Zhang, Wenhui Wu

**Affiliations:** 1Department of Marine Biopharmacology, College of Food Science and Technology, Shanghai Ocean University, Shanghai 201306, China; songtuo987@163.com (T.S.); xzdiao@shou.edu.cn (X.D.); d210300068@st.shou.edu.cn (Y.M.); 2234116@st.shou.edu.cn (B.C.); starfishhx@163.com (H.Z.); 2Solvo Biotherapeutics (Shanghai) Co., Ltd., Block 6, No.999 Huanke Road, Pudong New District, Shanghai 201210, China; juncheng23666@163.com; 3Putuo Branch of International Combined Research Center for Marine Biological Sciences, Zhoushan 316104, China

**Keywords:** plasminogen activator, *E. coli*, MBP, TEV, thrombolytic drug

## Abstract

As an essential thrombolytic agent, the tissue plasminogen activator receives increasing attention due to its longer half-life, lower immunogenicity, and easier administration, which are superior to other thrombolytic agents. In this study, the isolated and purified plasminogen activator from the sandworm (*Perinereis aibuhitensis*) was expressed in *E*. *coli* (*Escherichia coli*) to investigate its potential for simplifying the development process. The sandworm plasminogen activator was previously successfully cloned and expressed in *E*. *coli* with low yield and activity in the culture supernatant. This low yield and activity prompted us to optimize its DNA sequence. Furthermore, to raise the efficiency in the separation of the target protein, the protein’s solubility was enhanced by fusing it with maltose-binding protein (MBP) tags. Eventually, the fibrinolytic activity was successfully restored after digestion with tobacco etch virus (TEV) protease. This study provides an innovative method of efficiently expressing and purifying plasminogen activators from sandworm in *E*. *coli* and broadens its applications in therapeutic treatment of cardiovascular diseases, including thrombosis, stroke, and coronary atherosclerotic heart disease.

## 1. Introduction

Thrombotic diseases, including a range of serious health hazards such as myocardial infarction, ischemic stroke and pulmonary embolism, are usually triggered by intravascular thrombosis [1,2]. The blocked vessels during thrombosis consequently lead to severe pathological phenomena [3,4], which deserves equal attention to other cardiovascular diseases. When thrombolytic therapy is applied, the body’s fibrinolytic system is activated by the thrombolytic drugs to dissolve blood clots in the blood vessels and restore their patency [5,6]. The proteases derived from the sandworm (*Perinereis aibuhitensis*) with efficient and specific thrombolytic activity could be a promising thrombolytic compound for the development of novel thrombolytic agents [7,8].

Throughout the traditional preparation of plasminogen activator, repeating purification steps involved requires meticulous control to ensure the high purity and stability of the final product [9,10]. Although the plasminogen activator extracted from the sandworm (*Perinereis aibuhitensis*) showed a notable enzymatic activity, proved by our previous research [11,12], the supply instability of the sandworm related to breeding conditions, technology, and production rates limits its large-scale plasminogen activator preparation and application [13,14].

Serving as a widely utilized tool in protein engineering, the MBP tag is particularly pivotal in protein expression and purification [15,16], significantly enhances the solubility of fusion proteins in bacteria, and overcomes the challenges associated with poor protein solubility [17,18]. Addressing TEV protease, a cysteine protease derived from the tobacco etch virus with high sensitivity at the molecular level [19,20], is frequently employed to eliminate tags, such as MBP tags, from fusion proteins [21,22]. By incorporating TEV cleavage sites into fusion proteins, tags can be accurately removed through enzymatic cleavage in vitro, yielding pure target proteins [23,24]. In this study, the novel DNA sequence of plasminogen activator with His tag, MBP tag, and TEV protease were cloned and then expressed in *E. coli*. By this innovative approach, the preparation of plasminogen activator has been significantly simplified with improved efficiency [25,26]. This study would allow the amplified plasminogen activator to be produced from *E. coli* with high-quality through a straightforward fermentation and purification instead of the complicated extraction and purification steps [27,28].

## 2. Materials and Methods

### 2.1. Chemical and Reagents

#### 2.1.1. Reagents

LB (Kana) solid or liquid medium was made using yeast (OXOID, Basingstoke, Hampshire, UK, no. LP0021B). Tryptone (OXOID, Basingstoke, Hampshire, UK, no. LP0042); Sodium Chloride (Taitan, Shanghai, China, no. G81793F); Kanamycin Sulfate (Bio-engine, Shanghai, China, no. A600286-0025) with or without Agar (Bio-engine, Shanghai, China, no. A505255-0250). Elution solution and different concentrations of the washing buffer (pH8.0) were made using Imidazole (Aladdin, Darmstadt, Hessen, Germany, no. K2228466) (1.36, 2.72, 27.23 g/L, separately) in a 20 mM Tris-HCl solution. Since protein inactivation readily occured with a rapid change of the concentration, to slow down the process of denaturation and renaturation of inclusion bodies, the denaturing and renaturation solutions were made by using urea, Tris, and Tris-HCl, dissolved in 1 L deionized water, in different ratios. 1 mg/mL Plasmin(Sigma, Darmstadt, Hessen, Germany, no. MAX244C) was applied in this study. 10 × TEV Buffer was made by 500 mM Tris-HCl, 500 mM NaCl, 5 mM EDTA, and 10 mM DTT at pH 8.0. TEV protease (Novozymes, Biologiens, Lyngby, Denmark, no. JE1006-01), Coomassie Brilliant Blue G250 (Sangong, Shanghai, China, no. A100615-0005) was used for staining the protein after the SDS-PAGE.

Plasminogen (Shanghai Kuan-Dong Biological, Shanghai, China, no. 9001-91-6) was dissolved in 2 mL of sterile deionized water to make the plasminogen solution of 0.75 mg/mL. The solution underwent temperature equilibration before use, followed by dilution to the specific concentration.

Urokinase (Shanghai Guandong Bio, Shanghai, China, no. UPA005) was dissolved in 1 mL of sterile deionized water to make a urokinase solution of approximately 10,000 U/mL. After temperature equilibration, the solution was diluted to the specific concentration.

3.03 g of Tris and 4.38 g of NaCl were dissolved in 500 mL of deionized water. After the pH was adjusted to 7.4, the assay buffer was sterilized and stored at −20 °C, to be used later.

Fibrin analogues (AAT Bioquest, Sunnyvale, CA, USA, no. 13201) were diluted by assay buffer to 5 nmol/L.

Various buffer solutions were also prepared for western blotting, including a 20 mM PB (phosphate buffer) solution with 150 mM of NaCl at pH 7.2, 0.5 M of NaOH, a similar PB solution with 500 mM of Imidazole, 20% ethanol, a 20 mM PB solution with 2 M of NaCl, and a 1 M NaCl solution.

#### 2.1.2. Plasmid

pET28a(+) (TaKaRa. Kusatsu, Shiga, Japan); PDS418 (Nanjing Kingsley, Nanjing, China); CL4328 (synthesized by Nanjing Kingsley, pET28a(+) containing the target protein DNA before sequence optimization, schematic as in Appendix A); CL4329 (synthesized by Nanjing Kingsley, PDS418 containing the target protein DNA after sequence optimization, schematic as in Appendix A); expression host: BL21*(DE3) (Takara, TaKaRa. Kusatsu, Shiga, Japan).

### 2.2. Plasmid Transformation and Protein Expression

#### 2.2.1. Plasmid Transformation

BL21(DE3) receptor cells were removed from a –70 °C ultra-low temperature freezer and thawed on ice. The plasmid, containing no more than 100 ng of DNA, was introduced into a container, gently stirred by hand, and left to stand for 20 min. The contents were then placed in a 42 °C water bath, heated for 45 s, and immediately cooled on ice for 5 min. A mixture of 1 mL of LB medium was prepared on an ultra-clean bench (Suhui, SW-CJ-2FD), the temperature adjusted to 37 °C, and the mixture was oscillated at 200 rpm and incubated in a constant-temperature shock incubator (Shanghai Rundle Bioscience and Technology, Shanghai, China, ATec M1) continuously for 1 h. From the transformation, 200 µL was spread evenly onto the solid surface of LB (Kana) Agar using a sterile applicator stick and incubated at 37 °C. Subsequently, 300 µL of LB (Kana) liquid medium was added to a 48-well plate on an ultra-clean bench, and 10 single clones were selected for incubation at 37 °C at an oscillation speed of 200 rpm until an OD600 of 0.2–0.8, detected by Ultraviolet spectrophotometer (Shanghai Yuanyan Instrumentation, Shanghai, China, model V-5000), was achieved. Finally, 50 mL of LB (Kana) liquid medium was added to a sterile 250 mL flask on an ultra-clean workbench, inoculated with the bacterial solution, and incubated at 37 °C with 200 rpm oscillation until an OD600 of 1.0–1.5 was reached. The culture was then mixed with an equal volume of 30% glycerol, dispensed into freezing storage tubes, labeled, and stored at −60 °C.

#### 2.2.2. Validation of Shaking-Flask-Induced Expression

A measure of 200 mL of LB (Kana) liquid medium was added to a sterile 1000 mL flask within an ultra-clean workbench, and 10 mL of seed solution was inoculated, followed by incubation at 37 °C with 200 rpm oscillation. Inducer IPTG was then added to the medium at OD600 between 0.6 and 0.8, setting the final concentrations to 0.2 mM (after overnight incubation at 25 °C) and 0.8 mM (after overnight incubation at 37 °C), respectively. On a subsequent day, the bacterial solution was transferred into a centrifuge cup and centrifuged at 4 °C for 15 min at 12,000 rpm to collect the bacterial bodies. The collected bacteria were resuspended in 20 mM Tris-HCl pH 8.0 to a concentration of 25 mg/mL and subjected to a high-pressure homogenizer (Antos Nano Science and Technology, Suzhou, Jiangsu, China, AH-MINI) for homogenization and breaking at 4 °C under 800~900 bar for three cycles. One milliliter of the broken bacteria was then placed into a 1.5 mL centrifuge tube, centrifuged at 12,000 rpm at 4 °C for 10 min in a floor-standing high-speed freezing centrifuge (Hunan Kecheng Instrumentation Machinery, Changsha, Hunan, China, H6-10KR), and the upper layer of the supernatant was extracted and mixed with 1 mL of 20 mM Tris-HCl pH 8.0 (Table 1), before being transferred into the SDS-PAGE solution for further analysis.

#### 2.2.3. Inclusion Body Collection and Washing

Since the proteins mainly exist in the form of inclusion bodies in cells, the inclusion bodies were weighed after wall-breaking treatment on cell, and the inclusion body washing solution was added at a ratio of 1:10. The inclusion bodies were fully resuspended and centrifuged at 4 °C and 12,000 rpm for 30 min, after which the supernatant was removed; this step was repeated once.

#### 2.2.4. Inclusion Body Denaturation and Renaturation

A denaturing solution was added to the inclusion bodies, which was resuspended fully until all inclusion bodies were dissolved. Then, the whole solution was put in a dialysis bag and sequentially transferred into the renaturation solutions 1~4 with different urea concentrations, at 4 °C, and dialyzed for 8~16 h. After dialysis, the samples were taken to test enzyme activity directly, taking into consideration that the presence of urea in the renaturation solution may have had an effect on the enzyme activity.

### 2.3. Ni Column Gravity Column Purification

The Ni filler balancing liquid was washed until neutral, mixed with the sample, incubated, and sampled. The sample was then rinsed with an equilibrium solution and collected until there was no significant discoloration of the liquid under the Thomas Brilliant Blue reagent spot. After rinsing with Wash Solution 1, the sample was collected until there was no obvious discoloration of the liquid under Thomas blue reagent. Then, the samples were rinsed with Wash Solution 2 and collected until there was no obvious discoloration under Coomassie blue reagent. Finally, after elution, the above samples were subjected to SDS-PAGE, Western blotting, and enzyme activity tests.

### 2.4. Measurement of Enzyme Activity

#### 2.4.1. Urokinase Standard Curve

In each well of a 96-well enzyme plate, 2 µL of plasminogen and 2 µL of Fibrin analogue was added. A urokinase solution was diluted with deionized water in a gradient of 1 U/µL, 0.5 U/µL, 0.25 U/µL, 0.125 U/µL, and 0.0625 U/µL. Then, 1 µL of each solution was added to different wells of the 96-well enzyme plate, and the samples were immediately placed into the enzyme labeling instrument. The fluorescence was measured at 37 °C and read every 1 min for 30 min (Ex/Em = 360/450 nm), and a standard curve was generated with the fluorescence value at 30 min as the vertical coordinate and the urokinase activity as the horizontal coordinate (Table 2).

#### 2.4.2. Fibrinolytic Enzyme Activity Test

Fibrinolytic enzyme activity formulations for testing the fibrinolytic activity of the enzyme, as outlined in Table 3.

#### 2.4.3. Plasminogen Activator Enzyme Activity Test

Plasminogen activator enzyme activity formulations for testing the enzyme activity of the kinase, as outlined in Table 4.

#### 2.4.4. TEVase Digestion

The GenBank number of the resulting optimized amino acid sequence is AHZ01188.1 (Figure 1).

Plasminogen activator with an MBP tag was digested with TEVase according to the following ratios (Table 5): the digestion temperatures were 30 °C, and the digestion time was 1 h.

#### 2.4.5. Enzyme Marker Readings

The plate was placed into the Varioskan LUX enzyme labeling instrument (Thermo, 3020) immediately, the fluorescence was read under 37 °C at intervals of 1 min for 30 min (Ex/Em = 360/450 nm); then, the fluorescence value of the sample was calculated for the 30 min period. The plasminogen activator activity (U/µL) of the sample was calculated from the standard curve and fluorescence. The specific activity of plasminogen activator (U/mg) was calculated if the concentration of the sample protein was known.

### 2.5. Detection of Protein Amount by SDS-PAGE and Grayscale Scanning

#### 2.5.1. Loading and Electrophoresis

After adding 10 μL of the sample to the wells of the gel, electrophoresis was performed at 160 V for 50 min.

#### 2.5.2. Dyeing and Decolorization

After electrophoresis, the electrophoresis device was disassembled according to the installation process, and the rubber plate was removed. The prefabricated plate was detached from the side of the rubber plate with an iron skewer to expose the rubber block. A dyeing box was filled halfway with tap water, and the rubber block was gently transferred into the water. The water was poured out, the dyeing solution was added, and the glue block was submerged. The block was oscillated at 60 rpm on a flat plate oscillator for 60 min, and then, the dyeing solution was removed from the glue block. The dehydrating liquid was added and oscillated at 60 rpm on a flat plate oscillator. In the middle process, the decoloring solution was changed every 1 h until dehydration was completed. After decolorization, a gel imager was used to analyze the photographs.

#### 2.5.3. Imaging and Analysis

Imaging was performed using a gel imager. The gel was placed in the specified position of the imager, the imager power supply was turned on, and the imaging software AllDoc_x 6 was used. The images were analyzed manually by setting the exposure time to 20 ms and adjusting the focal length and aperture to make the image appear prominent. The key parameters of the image analysis were saved and are summarized in Table 6.

### 2.6. Western Blotting

After electrophoresis, the proteins are transferred from the gel to a membrane, polyvinylidene fluoride (PVDF), using a wet transfer method. Once on the membrane, the proteins are blocked to prevent non-specific binding, then incubated with primary antibody (Abcam, Cambridge, UK, no. ab62763) specific to the target protein, which binds to its epitope. After washing to remove unbound primary antibody, a secondary antibody(Abcam, Cambridge, UK, no. ab205718) linked to an enzyme or fluorescent tag is applied; this antibody recognizes and binds to the primary antibody. Following further washing, a substrate (Thermo, Carlsbad, CA, USA, no. 32209) for the enzyme is added, which upon enzyme-catalyzed reaction produces a visible signal, often a colored band or chemiluminescent signal, corresponding to the location of the target protein on the membrane. The specificity of the detection is confirmed by the unique molecular weight of the target protein band and the use of appropriate positive and negative controls.

## 3. Results

### 3.1. Isolation and Purification of the Plasminogen Activator from E. coli

The plasminogen activator expressed in *E. coli.*, and isolated by SDS-PAGE, significantly accumulated in the post-cell lysis precipitate via stimulation by 0.8 mM IPTG (isopropyl β-D-1-thiogalactopyranoside) at 37 °C, which indicated that the target protein was predominantly expressed in inclusion bodies under these conditions. However, the isolated plasminogen activator remarkably decreased with stimulation by 0.2 mM IPTG at 25 °C. Although the isolated plasminogen activator showed its presence in both the supernatant and the precipitate, it predominantly expressed as inclusion bodies in the precipitate under intense conditions (Figure 2).

The plasminogen activator-containing supernatant, achieved by the isolation from *E. coli* via stimulation by 0.2 mM IPTG at 25 °C, was then purified using the nickel column, followed by verification by both SDS-PAGE and Western blotting (WB) (Figure 3). The results by both SDS-PAGE and WB showed a target protein of 28.2 kD, which is consistent with our previously reported plasminogen activator from the sandworm (GenBank, ACL12061.1).

### 3.2. The Activation on Plasminogen of the Purified Plasminogen Activator

Upon evaluating the purified sample for enzyme activity, regrettably, no enzymatic activity was detected (Figure 4). This outcome suggests that either the original enzyme activity of the target protein was lost during purification or that impurities masked the enzyme activity due to low purity. To further investigate the underlying cause, we plan to meticulously review and analyze the purification process, aiming to pinpoint potential factors contributing to the loss of enzyme activity. Our goal is to refine the purification conditions or methods to potentially restore or enhance the enzyme activity of the target protein. Concurrently, we will explore alternative methods to verify the activity of the target protein, ensuring the accuracy and reliability of our experimental results.

The purified plasminogen activator exhibited no fibrinolytic activity but partly showed the activation of plasminogen (Figure 5), which indicated that the protein engaged in the fibrinolytic process by the initiation of plasmin-facilitating fibrinolysis instead of directly hydrolyzing fibrin.

The isolated plasminogen activator collected from the inclusion bodies by denaturation and renaturation also showed aofctivation on the plasminogen, whose effect was barely influenced by the purification of the protein via the fluid exchange (Figure 6). Given the fact that the plasminogen activator isolated from *E. coli* exclusively remained in the inclusion bodies with a minor fraction of the soluble one, the complex and time-consuming purification procedure of fluid exchange, increased the instability with an unavoidable substantial reducing effect in its activation.

### 3.3. Isolation and Purification of the Optimized Plasminogen Activator from E. coli

Compared to the plasminogen activator without optimization, the optimized plasminogen activator isolated from *E. coli* enriched in the supernatant after the cell lysis with soluble form under both two induction conditions (by 0.2 mM IPTG, at 25 °C or 0.8 mM IPTG, at 37 °C), which showed significant difference with the plasminogen activator without TEVase digestion (Figure 7a). The isolated optimized plasminogen activator showed the highest efficiency with ideal stability at the optimal induction conditions via stimulation by 0.2 mM IPTG at 25 °C, compared to the plasminogen activator before optimization (Figure 7b).

According to the results by SDS-PAGE, the purified optimized plasminogen activator (52.9 kDa) was gained by the nickel column without redundant bands, which indicated the remarkable efficacy of the purification of the target protein (Figure 8). Further quantitative analysis revealed that the concentration of the purified plasminogen activator is approximately 2 mg/mL, which provided high-quality samples for subsequent experiments.

### 3.4. The Activiation of Plasminogen of the Recombinant Plasminogen Activator after the TEVase Digestion

Same as the purified plasminogen activator without optimization, the purified optimized plasminogen activator also did not exhibit direct fibrinolytic activity of hydrolyzing the fibrin (Figure 9).

After TEVase was applied to digest the purified optimized plasminogen activator at the TEV site, the finally achieved recombinant plasminogen activator showed remarkable activation of the plasminogen (Figure 10), which indicated the digestion by TEVase not only eliminated the MBP tag, but also enabled the protein to regain and even improve its activation effect on the plasminogen.

## 4. Discussion

In this study, utilizing the plasmid of *E. coli* as the vector, the plasminogen activator originating from the sandworm (*Perinereis aibuhitensis*) was successfully expressed, isolated and purified. To respond to the challenges on the efficiency and cost once it is under large-scale production, which are mainly caused by the plasminogen activator being present predominantly in the inclusion bodies of the post-cell lysis precipitate, the optimization of the DNA sequence encoding for the target protein from the sandworm was applied. By incorporating a soluble MBP tag into the DNA sequence, inclusion body formation was inhibited resulting with an increase in solubility of the target protein, which consequently increased the optimized plasminogen activator in the supernatant, circumventing the complex process on the inclusion body. The sandwich-type sequence of the optimized plasminogen activator significantly improved the protein’s solubility and enhanced the efficiency of the synthesis in *E. coli*. As a result, the amount of the optimized plasminogen activator dramatically increased in the supernatant, which benefited the subsequent purification. Importantly, the solid–liquid separation could be simply achieved via centrifugation, allowing for the direct isolation and purification of the target protein from the supernatant, bypassing the process of inclusion body treatment, and helping to preserve the integrity and activity of the protein. It is the first report on the application of MBP tag in the isolation and purification of the plasminogen activator from sandworm.

However, the insert of the MBP tag unavoidably influenced the activation effect on the plasminogen of the purified optimized plasminogen activator. Therefore, the cleavage site of TEV was further investigated based on prior research. It has been eventually discovered that through the synergistic action of MBP and TEV, the limitation could be overcome. When the TEVase was employed to digest the purified optimized plasminogen activator through the TEV site, the MBP tag initially attached to the protein was accurately excised, which consequently enabled the recombinant plasminogen activator to fully exhibit its enzymatic properties. After TEVase digestion, the obstruction of the activation on the plasminogen of the recombinant plasminogen activator was overcome with even remarkably increasing activity, which also demonstrated that the addition of the TEV sequence did not adversely affect the activity of the sandworm plasminogen activator. This study introduced a novel purification method of the plasminogen activator from the sandworm. The advantages of the optimized purification method are shown in Table 7.

Notably, different from the traditional fibrinolytic reagents, the recombinant plasminogen activator exhibits its novel fibrinolytic activity by the activation on the plasminogen instead of directly hydrolyzing the fibrin.

## 5. Conclusions

By optimizing the sequence, the soluble plasminogen activator was successfully expressed in recombinant *E. coli*. After the TEVase digestion, the purified optimized plasminogen activator gained by the nickel column exhibited fibrinolytic activity via the activation on the plasminogen. This modification suggests that the protein no longer requires a refolding process to regain its native structure and function. The improved solubility probably results from the addition of the amino acid residues that promote hydrophilic interactions, thereby facilitating the enzyme’s stability and activity in aqueous solutions without the need for the additional renaturation steps. This study provides an innovative method of efficiently expressing and purifying plasminogen activator from the sandworm in *E. coli* and broadens its applications in therapeutic treatment in cardiovascular diseases, including thrombosis, stroke, and coronary atherosclerotic heart disease.

## Figures and Tables

**Figure 1 bioengineering-11-01030-f001:**
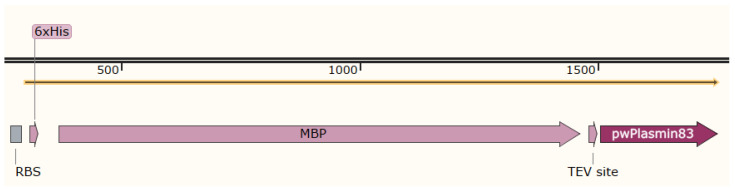
Schematic representation of the complete protein structure to be expressed (MBP is the maltose-binding protein tag, TEV is the TEV cleavage site of the protease, and pwPlasmin83 is the 83AA to be expressed).

**Figure 2 bioengineering-11-01030-f002:**
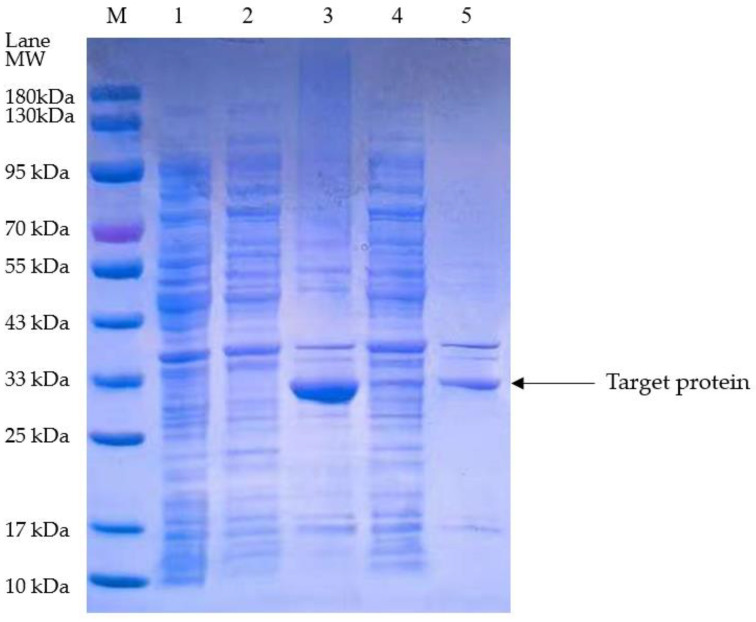
SDS-PAGE electrophoretic graph. M: Marker; Lane 1: without IPTG induced; 2: supernatant, the temperature reaches 37 °C, while the concentration of IPTG is 0.8 mM; 3: precipitation, the temperature reaches 37 °C, while the concentration of IPTG is 0.8 mM; 4: supernatant, the temperature reaches 25 °C while the concentration of IPTG is 0.2 mM; 5: precipitation, the temperature reaches 25 °C while the concentration of IPTG is 0.2 mM.

**Figure 3 bioengineering-11-01030-f003:**
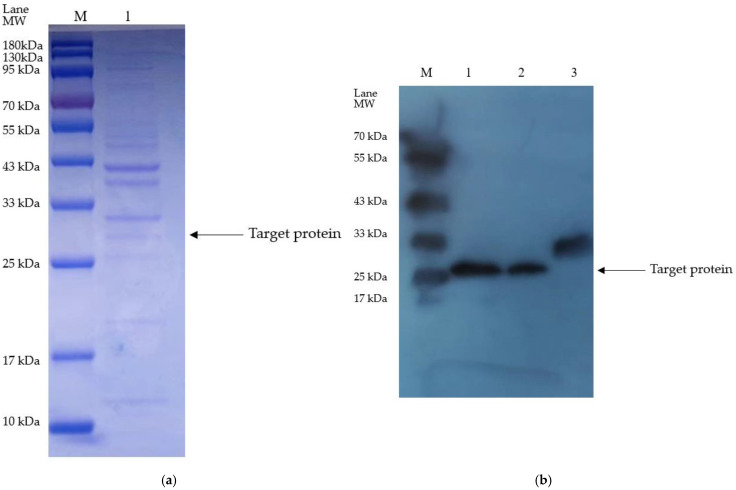
(**a**) Electrophoretic graph, M: marker, Lane1: elution. (**b**) WB graph, Lanes 1 and 2: same elution after nickel column purification, 3: control (PNGase F with His Tag).

**Figure 4 bioengineering-11-01030-f004:**
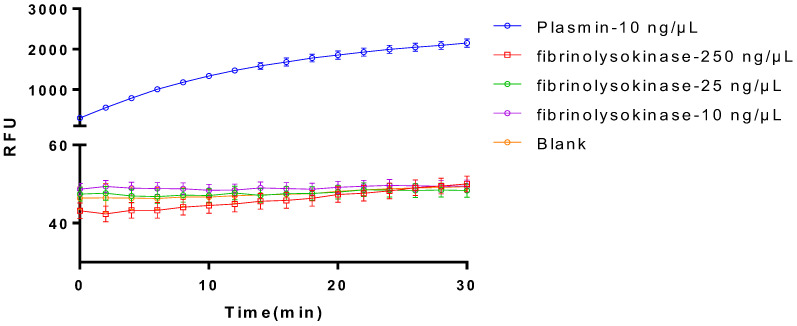
Comparison on the fibrinolytic activities between plasmin and the purified plasminogen activator. The data represent mean ± SD, *n* = 3.

**Figure 5 bioengineering-11-01030-f005:**
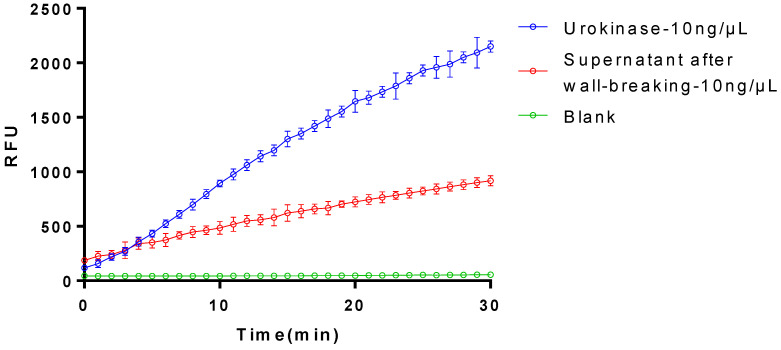
Comparison on the activation of plasminogen between urokinase and the purified plasminogen activator. Supernatant after wall-breaking refers to the liquid that remains on top after the cell walls have been broken or lysed. The data represent mean ± SD, *n* = 3.

**Figure 6 bioengineering-11-01030-f006:**
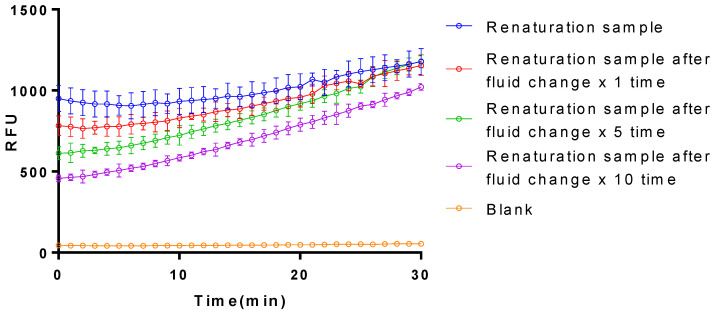
Influence on the activation of plasminogen by purification via denaturation of the isolated plasminogen activator in the inclusion body from *E. coli*. the sample that has been refolded is then referred to as “Renaturation sample”. Renaturation sample after fluid change ×1 time: the sample after renaturation that has undergone one single fluid change treatment. Renaturation sample after fluid change × 5 time: the sample after reconstitution that has undergone 5 rounds of fluid change treatment. Renaturation sample after fluid change × 10 time: the sample after reconstitution that has undergone 10 rounds of fluid change treatment. The data represent mean ± SD, *n* = 3.

**Figure 7 bioengineering-11-01030-f007:**
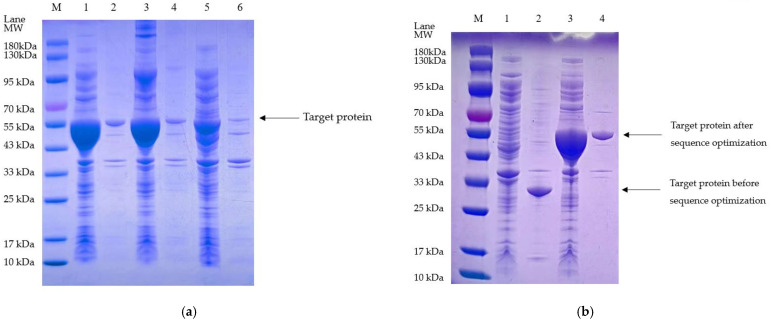
(**a**) Electropherogram after sequence optimization. M: marker. Lane1: supernatant, the temperature reaches 25 °C while the concentration of IPTG is 0.2 mM; 2: precipitation, the temperature reaches 25 °C while the concentration of IPTG is 0.2 mM; 3: supernatant, the temperature reaches 37 °C while the concentration of IPTG is 0.8 mM; 4: precipitation, the temperature reaches 37 °C while the concentration of IPTG is 0.8 mM; 5: supernatant, without the induction of IPTG; 6: precipitation, without the induction of IPTG. (**b**) Comparison of electropherograms before and after sequence optimization; Lane1: before optimization, supernatant; 2: before optimization, precipitation; 3: after optimization, supernatant; 4: after optimization, precipitation.

**Figure 8 bioengineering-11-01030-f008:**
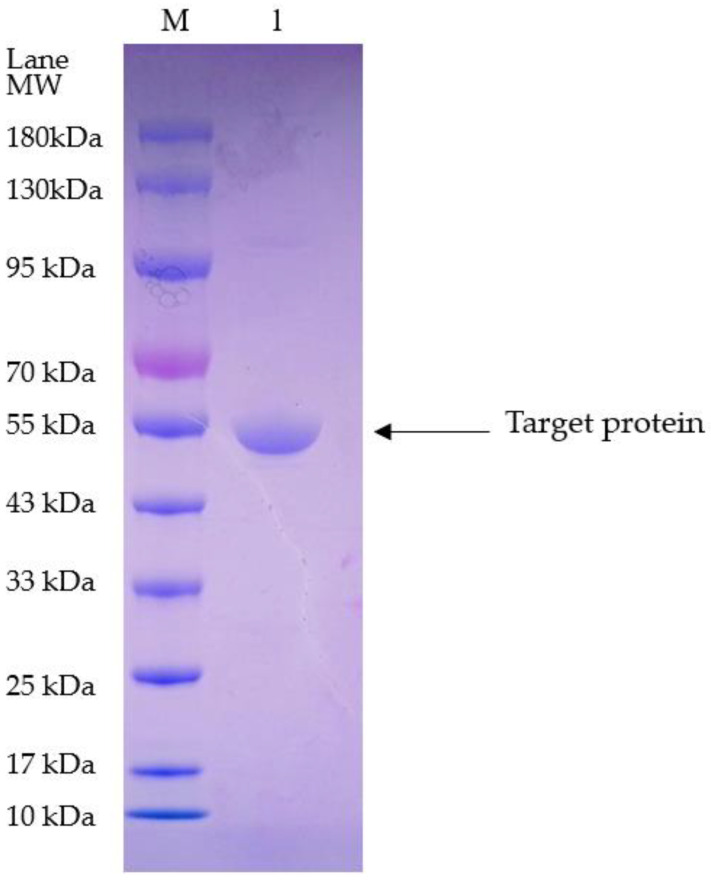
SDS-PAGE electrophoretic graph. M: marker; Lane1: elution.

**Figure 9 bioengineering-11-01030-f009:**
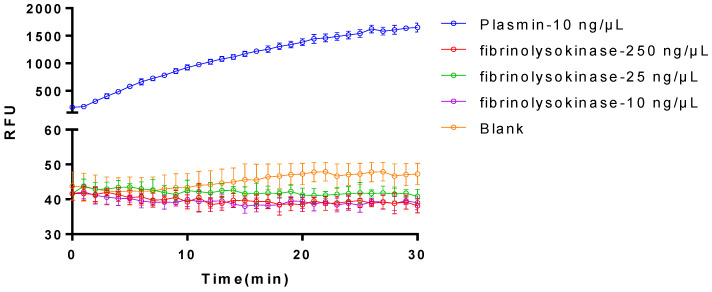
Fibrinolytic protease enzyme activity assay. The data represent mean ± SD, *n* = 3.

**Figure 10 bioengineering-11-01030-f010:**
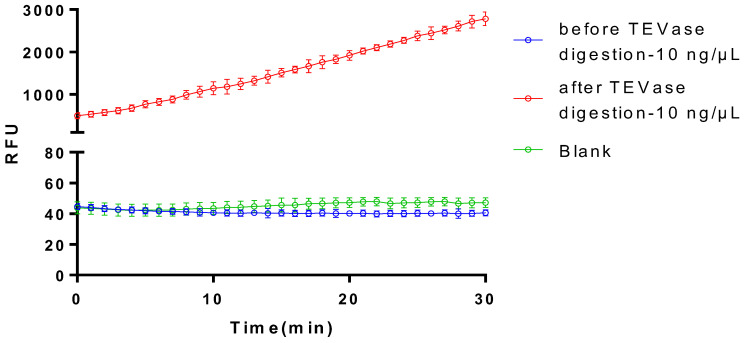
Determination of the enzyme activity of plasminogen activator before and after cleavage. Before TEVase digestion: samples before being treated with TEV enzymes; after TEVase digestion: samples treated with TEV enzymes. The data represent mean ± SD, *n* = 3.

**Table 1 bioengineering-11-01030-t001:** Steps for bacterial cells wall broken.

Description	Note
Diluent	20 mM Tris-HCl, pH 7.2
	1:10 (bacteria: dilution solution (*w*/*w*)) for 30 min.
Homogenize	Bale breaking with high-pressure homogenizers
Centrifuge after breaking cell wall	12,000 rpm, 10 min
Post centrifugation filtration	0.8 µm-pore-size filter paper filtration
Sample adjustment	The conductance of the homogenate was adjusted to 15–17 mS/cm using 1 M NaCl

**Table 2 bioengineering-11-01030-t002:** Linear formulations.

Reagent	Volume				
Assay Buffer	48 µL				
Fibrin analogues	2 µL				
Plasminogen	2 µL				
Urokinase	0.0625 U/µL	0.125 U/µL	0.25 U/µL	0.5 U/µL	1 U/µL
Assay Buffer	to 100 µL				

**Table 3 bioengineering-11-01030-t003:** Fibrinolytic enzyme activity test formulation.

Reagent	Volume			
	Plasmin (10 ng/µL)	Plasminogen Activator (250 ng/μL)	Plasminogen Activator (25 ng/μL)	Plasminogen Activator (10 ng/μL)
Assay Buffer	48 µL			
Fibrin analogues	2 µL			
Enzyme	25 µL	25 µL	25 µL	25 µL
Assay Buffer	to 100 µL			

**Table 4 bioengineering-11-01030-t004:** Plasminogen activator enzyme activity test formulation.

Reagent	Volume	
	Samples	Positive Reference Material
Assay Buffer	48 µL	
Fibrin analogues	2 µL	
Plasminogen	2 µL	2 µL
Urokinase (10 ng/µL)	N/A	25 µL
Plasminogen activator (10 ng/µL)	25 µL	N/A
Assay Buffer	to 100 µL	

**Table 5 bioengineering-11-01030-t005:** TEVase digestion system.

Individual Parts Making Up a Compound	Volumetric
TEV enzyme (10 U/µL)	1 µL
sample protein	30 µg
10 × TEV Buffer	To 50 µL

**Table 6 bioengineering-11-01030-t006:** Key parameters of SDS-PAGE image analysis.

Parameter Name	Set Value
Strip sensitivity	0.50–5.0 (can be adjusted according to the actual situation; ensure that all visible strips are identified and only a small number of strip identifiers in the blank lane is present ≤ 5)
Select background mode	Peak-valley connection
Baseline connections	1% (can be adjusted according to the actual situation so that the baseline is level, and the background is completely deducted)
Molecular weight regression model	Exponential regression
Quantitative approach	Lane total/relative amount (tick)
Results report form	The “relative quantity” and “IOD” forms are saved separately.
Protein amount Results	The IOD value is the gray value

**Table 7 bioengineering-11-01030-t007:** Performance comparison of purification steps.

	Before Sequence Optimized	After Sequence Optimized
SDS-PAGE Results	The target protein bands were not obvious	After adding DNA sequence to MBP tag: the target protein bands are clear and single
Protein concentration	2 mg/mL	2 mg/mL
Total protein	330 mg	1.2 g
Plasminogen activator is active	3 U/μL	15 U/μL
Yield	10%	85%

## Data Availability

The data that support the findings of this study are available on request from the corresponding author.

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
