# Peer review of "Recombinant Plasminogen Activator of the Sandworm (Perinereis aibuhitensis) Expression in Escherichia coli"

_bioengineering, 2024, doi:10.3390/bioengineering11101030_

Round 1

Reviewer 1 Report

Comments and Suggestions for Authors

Dear editor in chief,

The manuscript, “Recombinant fibrinolysokinase of the sandworm (Perinereis aibuhitensis) expression in Escherichia coli,” is well-designed; however, some flaws must be properly addressed.

How the construct was prepared?

Did they extract RNA from the sandworm and, after RT-PCR, ligate and transform it, or did they design and purchase the construct containing the target sequence of the target enzyme of the worm? Please clarify it in the text. If the construct used from their previous work, mention it with the reference.

They addressed DNA; since it is a coding region, did they use the specific exon or whole?

There was no primer or PCR condition, so please clarify how to prepare the construct.

The Ab was not mentioned in Western blotting, e.g., the dilution and the source, anti-goat or anti-rabbit IgG.

How did they determine the purity of protein? FPLC or HPLC?

Line 20, Which

Line 106, Various buffer solutions for which purpose?

Line 106: the full name of PB should be mentioned.

Line 161: Please rewrite this sentence to avoid misunderstanding. The protein before optimization, because of its own amino acid sequence, is a protein in the cell in large quantities in the form of inclusion bodies; therefore, after wall-breaking treatment, additional inclusion bodies need to be added.

Figure 3b, in the caption, each lane should be explained. 

The text should be edited. 

Comments on the Quality of English Language

The text should be edited. There were some grammatical errors.

Author Response

List of Actions

LOA1: “fibrinolysokinase” has been replaced by “plasminogen activator” in the Title, Abstract, Keywords and throughout the whole Text. (line 2,14-27,39-44,53-57,77,78,161,169-181,192-193,231-244,257,267-274,280-284,287-306,322-331,335-384)

LOA2: Reference 1-2 have been added in the Abstract. (line 15)

LOA3: “which”has been replaced by “such low yield and activity” in the Abstract to make the expression clearer. (line 20)

LOA4: “Furthermore” has been corrected in the Abstract. (line 20)

LOA5: “therapeutic” has been corrected in the Abstract. (line 25)

LOA6: “ischemic” has been corrected in the Introduction. (line 31)

LOA7: “promising” has been corrected in the Introduction. (line 37)

LOA8: “stability” has been corrected in the Introduction. (line 40)

LOA9: The 2.1. Chemical and Reagents has been re-written to give more precise and clear expression. (line 63-101)

LOA10: The purpose of why different concentrations of solutions were prepared has been given in the 2.1.1. Reagents. (line 69-70)

LOA11: The role of Various buffer solutions and the full name of the PB has been added to the 2.1.1. Reagents (line89,90)

LOA12: The misspelling of the company name of Ultraviolet spectrophotometer has been corrected in the 2.2.1. Plasmid Transformation. (line 116)

LOA13: New contents about the Plasmid information have been added as the 2.1.2. Plasmid. (line 96-101)

LOA14: The comma has been added in the “12000 rpm” in the 2.2.2. Validation of Shaking-Flask-Induced Expression and 2.2.3. Inclusion Body Collection and Washing. (line 129,134,143)

LOA15: The misspelling of “Post centrifugation” has been corrected in the Table 1. (Table 1)

LOA16: The sentence “The protein before optimization…....additional inclusion bodies need to be added.” was optimized in 2.2.3. Inclusion Body Collection and Washing. (line 140-141)

LOA17: “a” was corrected as “the” in 2.2.3. Inclusion Body Collection and Washing. (line 142)

LOA18: “dialyze” was corrected in 2.2.4. Inclusion Body Denaturation and Renaturation. (line 148)

LOA19: The capitalization on “plasminogen” has been removed in the 2.4.1. Urokinase standard curve. (line 163)

LOA20: The concentrations of the fibrin analogues have been corrected in the 2.4.1. Urokinase standard curve. (line 164)

LOA21: The bracket of “Table 2” has been fulfilled in 2.4.1. Urokinase standard curve. (line 170)

LOA22: The item name has been capitalized in the Table 5. (Table 5)

LOA23: “immediately” has been corrected in the 2.4.5. Enzyme marker readings. (line 192)

LOA24: The 2.6. Western blotting has been re-written to give more precise and clear expression. (line 220-231)

LOA25: The full-spelling of “IPTG” has been given in the 3.1. Isolation and Purification of the Plasminogen activator from E. coli. (line 236)

LOA26: The caption of Figure 3b has been optimized. (line 255)

LOA27: “activation” has been corrected in the title of the 3.2. The Activation on Plasminogen of the Purified Plasminogen Activator. (line 257)

LOA28: “exclusively” has been corrected in the 3.2. The Activation on Plasminogen of the Purified Plasminogen Activator. (line 276)

LOA29: “activative” has been replaced by “activation” in the 3.2. The Activation on Plasminogen of the Purified Plasminogen Activator and 3.4. The Activiation on Plasminogen of the Recombinant Plasminogen activator after the TEVase Digestion. (line 270,274,279)

LOA30: “comparison” has been corrected in the title of Figure 4 and Figure 5. (line 282, 285)

LOA31: “treatment” has been corrected in the caption of Figure 6. (line 292, 294, 295)

LOA32: “optimized” has been corrected in the title and the content of the 3.3. Isolation and Purification of the Optimized Plasminogen activator from E. coli and 3.4. The Activation on Plasminogen of the Recombinant Plasminogen activator after the TEVase Digestion. (line 297, 304)

LOA33: “therapeutic” has been corrected in the Conclusion. (line 387)

LOA34: “Ethics Approval Statement” has been added. (line 405-407)

LOA35: Several references have been added as Reference1-2, 12-13, and all the references have been recorded with the format being standardized.(Reference)

LOA36: xiaozhen Diao was changed to Co-first author due to her contributions both in the preparation and revision of the manuscript.(line 4)

[To Reviewer-1]

Dear reviewer:

Thank you for your helpful suggestions on the details of the materials and methods essential for the molecular biology, which have been added to the text and the Supplementary Materials, respectively. Moreover, according to your helpful suggestions, expression has been re-organized to make the more clearly defined results. The manuscript has been carefully revised depending on your thoughtful comments as follows. Thank you for all your precious suggestions to help improve this manuscript.

Comments 1:[How the construct was prepared? Did they extract RNA from the sandworm and, after RT-PCR, ligate and transform it, or did they design and purchase the construct containing the target sequence of the target enzyme of the worm? Please clarify it in the text. If the construct used from their previous work, mention it with the reference.]

Respond 1: [The process of sequence synthesis and plasmid extraction has been outsourced to the Kingsley Company. The information of the COA and synthesized plasmids before and after the optmization has been provided in the Supplementary Material 1 & 2.]

Comments 2:[They addressed DNA; since it is a coding region, did they use the specific exon or whole? There was no primer or PCR condition, so please clarify how to prepare the construct. The Ab was not mentioned in Western blotting, e.g., the dilution and the source, anti-goat or anti-rabbit IgG.]

Respond 2: [The process of sequence synthesis and plasmid extraction has been outsourced to Kingsley Company. I will only receive COA and synthesized plasmids here, please check Supplementary Material1 and 2.]

Comments 3:[ How did they determine the purity of protein? FPLC or HPLC?]

Respond 3: [In fact, the Kingsley Company has tested the purity of the plasmid instead of the protein; whose information would be kept private from the public.]

Comments 4:[Line 20, Which.]

Respond 4: [The sentence has been re-written to make it more clearly presented, see LOA3.]

Comments 5:[Line 106, Various buffer solutions for which purpose?]

Respond 5: [Proteins tend to lose their activity by the rapid change from high concentration to the low during the denaturation and renaturation, in which case, solutions with concentration gradient were prepared and applied to avoid the potential protein inactivation, see LOA10.]

Comments 6:[Line 106: the full name of PB should be mentioned.]

Respond 6:[These buffers are used in WB experiments and the full name of PB is phosphate buffer. see LOA11]

Comments 7:[ Line 161: Please rewrite this sentence to avoid misunderstanding. The protein before optimization, because of its own amino acid sequence, is a protein in the cell in large quantities in the form of inclusion bodies; therefore, after wall-breaking treatment, additional inclusion bodies need to be added.]

Respond 7: [As you kindly suggested, the expression has been optimized, see LOA15.]

Comments 8:[ Figure 3b, in the caption, each lane should be explained.]

Respond 8: [The explanation of Lanes has been given in the caption of Figure 3b, see LOA25.]

Reviewer 2 Report

Comments and Suggestions for Authors

This paper reports the expression of a recombinant enzyme “fibrinolysokinase” from the sandworm (Perinereis aibuhitensis) in E. coli.  Prior work by the group on isolation of the enzyme from the sandworm apparently showed notable fibrinolytic activity.  Purified after preparing the enzyme in bacteria with 0.2 mM IPTG at 25oC, the species failed to show activity against a fluorescently labeled fibrin analogue substrate for plasmin.  Lysis was demonstrated however in the presence of plasminogen.  Stimulation with 0.8 mM IPTG at 37oC yielded significantly more product in the form of inclusion bodies which cleaved plasminogen after denaturation-renaturation steps.   The process was optimized through production of a fusion protein containing maltose binding protein for improved solubility.  Treatment with tobacco etch virus protease (TEV) restored activity against plasminogen. 

Thrombolytic therapy saves the lives of individuals with acute myocardial infarction and stroke so the enzyme produced could potentially be a valuable agent for intervention.  If correct, the first statement in the abstract would support the importance of this contribution.  It is doubtful, however, that this enzyme is less immunogenic than tissue plasminogen activator and a literature search for “fibrinolysokinase” fails to produce prior work suggesting a longer half-life and easier administration. This sentence must be removed or substantiated in the Introduction with references to archival publications.

The term “fibrinolysokinase” is dated, appearing in articles 60-70 years ago, and needs to be replaced with another designation.  They have shown and can conclude that the sand worm enzyme is in fact a plasminogen activator.  Plasminogen activator and fibrinolysis would be good choices for your key words.  

Mention is made in the Introduction of their previous research (line 42).  References 11 and 12 at the end of the sentence do not lead the reader to their prior work nor do any of the other references do so.  Provide references to your earlier studies.  Authors might also include a paper by Zhao and Ju (Cloning, expression and activity analysis of a novel fibrinolytic serine protease from Arenicola cristata. Journal of Ocean University of China. 2015 Jun;14:533-40). 

Figure 9 can be eliminated as it has already been demonstrated in the paper that the enzyme is a plasminogen activator and does not directly cause fibrinolysis.

There are many typographical errors that need to be addressed.   The style in some of the references is wrong.

Plasmin seems to have been omitted in the list of reagents.

Line 122: What was the concentration of the fibrin analogues?

Line 187: …0.5 U/μL, 2.5 U/μL, 0.125 U/μL…         Do you mean 5.0 U/μL?

Comments on the Quality of English Language

There are many typographical errors and some odd wording of sentences.

Author Response

List of Actions

LOA1: “fibrinolysokinase” has been replaced by “plasminogen activator” in the Title, Abstract, Keywords and throughout the whole Text. (line 2,14-27,39-44,53-57,77,78,161,169-181,192-193,231-244,257,267-274,280-284,287-306,322-331,335-384)

LOA2: Reference 1-2 have been added in the Abstract. (line 15)

LOA3: “which”has been replaced by “such low yield and activity” in the Abstract to make the expression clearer. (line 20)

LOA4: “Furthermore” has been corrected in the Abstract. (line 20)

LOA5: “therapeutic” has been corrected in the Abstract. (line 25)

LOA6: “ischemic” has been corrected in the Introduction. (line 31)

LOA7: “promising” has been corrected in the Introduction. (line 37)

LOA8: “stability” has been corrected in the Introduction. (line 40)

LOA9: The 2.1. Chemical and Reagents has been re-written to give more precise and clear expression. (line 63-101)

LOA10: The purpose of why different concentrations of solutions were prepared has been given in the 2.1.1. Reagents. (line 69-70)

LOA11: The role of Various buffer solutions and the full name of the PB has been added to the 2.1.1. Reagents (line89,90)

LOA12: The misspelling of the company name of Ultraviolet spectrophotometer has been corrected in the 2.2.1. Plasmid Transformation. (line 116)

LOA13: New contents about the Plasmid information have been added as the 2.1.2. Plasmid. (line 96-101)

LOA14: The comma has been added in the “12000 rpm” in the 2.2.2. Validation of Shaking-Flask-Induced Expression and 2.2.3. Inclusion Body Collection and Washing. (line 129,134,143)

LOA15: The misspelling of “Post centrifugation” has been corrected in the Table 1. (Table 1)

LOA16: The sentence “The protein before optimization…....additional inclusion bodies need to be added.” was optimized in 2.2.3. Inclusion Body Collection and Washing. (line 140-141)

LOA17: “a” was corrected as “the” in 2.2.3. Inclusion Body Collection and Washing. (line 142)

LOA18: “dialyze” was corrected in 2.2.4. Inclusion Body Denaturation and Renaturation. (line 148)

LOA19: The capitalization on “plasminogen” has been removed in the 2.4.1. Urokinase standard curve. (line 163)

LOA20: The concentrations of the fibrin analogues have been corrected in the 2.4.1. Urokinase standard curve. (line 164)

LOA21: The bracket of “Table 2” has been fulfilled in 2.4.1. Urokinase standard curve. (line 170)

LOA22: The item name has been capitalized in the Table 5. (Table 5)

LOA23: “immediately” has been corrected in the 2.4.5. Enzyme marker readings. (line 192)

LOA24: The 2.6. Western blotting has been re-written to give more precise and clear expression. (line 220-231)

LOA25: The full-spelling of “IPTG” has been given in the 3.1. Isolation and Purification of the Plasminogen activator from E. coli. (line 236)

LOA26: The caption of Figure 3b has been optimized. (line 255)

LOA27: “activation” has been corrected in the title of the 3.2. The Activation on Plasminogen of the Purified Plasminogen Activator. (line 257)

LOA28: “exclusively” has been corrected in the 3.2. The Activation on Plasminogen of the Purified Plasminogen Activator. (line 276)

LOA29: “activative” has been replaced by “activation” in the 3.2. The Activation on Plasminogen of the Purified Plasminogen Activator and 3.4. The Activiation on Plasminogen of the Recombinant Plasminogen activator after the TEVase Digestion. (line 270,274,279)

LOA30: “comparison” has been corrected in the title of Figure 4 and Figure 5. (line 282, 285)

LOA31: “treatment” has been corrected in the caption of Figure 6. (line 292, 294, 295)

LOA32: “optimized” has been corrected in the title and the content of the 3.3. Isolation and Purification of the Optimized Plasminogen activator from E. coli and 3.4. The Activation on Plasminogen of the Recombinant Plasminogen activator after the TEVase Digestion. (line 297, 304)

LOA33: “therapeutic” has been corrected in the Conclusion. (line 387)

LOA34: “Ethics Approval Statement” has been added. (line 405-407)

LOA35: Several references have been added as Reference1-2, 12-13, and all the references have been recorded with the format being standardized.(Reference)

LOA36: xiaozhen Diao was changed to Co-first author due to her contributions both in the preparation and revision of the manuscript.(line 4)

[To Reviewer-2]

Dear reviewer:

We appreciate your affirmation on this study and thank you for your critical review. In all, the manuscript has been carefully revised depending on your thoughtful comments as follows. Thank you for all your precious suggestions to help improve this manuscript.

Comments 1:[Thrombolytic therapy saves the lives of individuals with acute myocardial infarction and stroke so the enzyme produced could potentially be a valuable agent for intervention.  If correct, the first statement in the abstract would support the importance of this contribution.  It is doubtful, however, that this enzyme is less immunogenic than tissue plasminogen activator and a literature search for “fibrinolysokinase” fails to produce prior work suggesting a longer half-life and easier administration. This sentence must be removed or substantiated in the Introduction with references to archival publications.]

Respond 1: [As suggested, relevant literature has been added to the abstract, see LOA2, 34.]

Comments 2:[ The term “fibrinolysokinase” is dated, appearing in articles 60-70 years ago, and needs to be replaced with another designation.  They have shown and can conclude that the sand worm enzyme is in fact a plasminogen activator.  Plasminogen activator and fibrinolysis would be good choices for your key words.  ]

Respond 2: [Thank you for your critical suggestion on the expression of the target enzyme of this study and all the “fibrinolysokinase” in both the title and the text have been replaced by “plasminogen activator”, see LOA1.]

Comments 3:[Mention is made in the Introduction of their previous research (line 42).  References 11 and 12 at the end of the sentence do not lead the reader to their prior work nor do any of the other references do so.  Provide references to your earlier studies.  Authors might also include a paper by Zhao and Ju (Cloning, expression and activity analysis of a novel fibrinolytic serine protease from Arenicola cristata. Journal of Ocean University of China. 2015 Jun;14:533-40).]

Respond 3: [Thank you for your thoughtful suggestions, and the relevant references have been added to the Introduction, see LOA 34]

Comments 4:[ Figure 9 can be eliminated as it has already been demonstrated in the paper that the enzyme is a plasminogen activator and does not directly cause fibrinolysis.]

Respond 4: [Although the optimized enzyme has been proved to show no direct fibrinolytic effect after the whole work of this study, it was uncertain whether it had direct fibrinolytic activity or not right after we optimized the sequence. In this case, Figure 9 would provide the evidence for the opinion.]

Comments 5:[ There are many typographical errors that need to be addressed.]

Respond 5: [We sincerely apologize for our imprudent miss spellings, and all the mistakenly spelled phrases and words have been corrected, see LOA 4-8, 12-14, 16-18, 20-22, 25-32.]

Comments 6:[The style in some of the references is wrong.]

Respond 6: [The format of all references has been standardized in the References, see LOA34.]

Comments 7:[Plasmin seems to have been omitted in the list of reagents.]

Respond 7: [The 2.1.1. Reagents has been re-written with the information of the plasmin added, see LOA9.]

Comments 8:[Line 122: What was the concentration of the fibrin analogues?]

Respond 8: [The information of the fibrin analogues has been added in the 2.1.1. Reagents, see LOA9.]

Comments 9:[Line 187: …0.5 U/μL, 2.5 U/μL, 0.125 U/μL…Do you mean 5.0 U/μL?]

Respond 9: [As your kindly mentioned, the concentrations of the fibrin analogues have been corrected, see LOA19.]

Round 2

Reviewer 2 Report

Comments and Suggestions for Authors

In the first sentence of the abstract, replace "plasminogen activator" with "tissue plasminogen activator" and remove references 1 and 2.  It has not been established that the plasminogen activator from sandworm has these properties.

Comments on the Quality of English Language

The English grammar is satisfactory.   You should review your manuscript to be sure the wording of a sentence makes clear when you are referring to the plasminogen activator from sandworm as opposed to when you are referring to other plasminogen activators.

Author Response

List of Actions

LOA1: “tissue” has been added in Abstract. (line 14)

LOA2: Reference 1-2 has been deleted from the Abstract and Reference.

[To Reviewer]

Dear reviewer:

Thank you for your helpful suggestions and the manuscript has been carefully revised depending on your thoughtful comments as follows. Thank you for all your help with improving this manuscript.

Comments 1:[ In the first sentence of the abstract, replace "plasminogen activator" with "tissue plasminogen activator".]

Respond 1: [As your helpful suggestion, the expression of "plasminogen activator" has been optimized as "tissue plasminogen activator ", see LOA1]

Comments 2:[ Remove references 1 and 2.]

Respond 2: [As your suggestion, References 1&2 have removed, see LOA2]

Comments 3:[ You should review your manuscript to be sure the wording of a sentence makes clear when you are referring to the plasminogen activator from sandworm as opposed to when you are referring to other plasminogen activators.]

Respond 3: [As your kindly reminding, only Urokinase has been used and was mentioned in the manuscript, in which case it would not cause the confusion with our plasminogen activator from sandworm.]